



# Hourly historical and near-future weather and climate variables for energy system modelling

Hannah C. Bloomfield[1,2], David J. Brayshaw[1,3], Matthew Deakin[4], and David Greenwood[4]

[1]Department of Meteorology, University of Reading (UK)
[2]School of Geographical Sciences, University of Bristol (UK)
[3]National Centre for Atmospheric Science, Reading (UK)
[4]Newcastle University, Newcastle-upon-Tyne, UK

**Correspondence:** h.c.bloomfield@reading.ac.uk

**Abstract.**

Energy systems are becoming increasingly exposed to the impacts of weather and climate due to the uptake of renewable generation and the electrification of the heat and transport sectors. The need for high-quality meteorological data to manage present and near-future risks is urgent. This paper provides a comprehensive set of multi-decadal, time series of hourly meteorological variables and weather-dependent power systems components for use in the energy systems modelling community. Despite the growing interest in the impacts of climate variability and climate change on energy systems over the last decade, it remains rare for multi-decadal simulations of meteorological data to be used within detailed simulations. This is partly due to computational constraints, but also due to technical barriers limiting the use of meteorological data by non-specialists. This paper presents a new European level dataset which can be used to investigate the impacts of climate variability and climate change on multiple aspects of near-future energy systems. The datasets correspond to a suite of well-documented, easy-to-use, self-consistent hourly- nationally-aggregated and sub-national time series for 2m temperature, 10m wind speed, 100m wind speed, surface solar irradiance, wind power capacity factor, solar power factor and degree days spanning over 30 European countries. This dataset is available for the historical period (1950-2020), and is accessible from https://researchdata.reading.ac.uk/id/eprint/321with reserved DOI: http://dx.doi.org/10.17864/1947.000321 (Bloomfield and Brayshaw, 2021b).

As well as this a companion dataset is created where the ERA5 reanalysis is adjusted to represent the impacts of near-term climate change (centred on the year 2035) based on five high resolution climate model simulations. This data is available for a 70 year period for central and Northern Europe. The data is accessible from https://researchdata.reading.ac.uk/id/eprint/331 with reserved DOI: http://dx.doi.org/10.17864/1947.000331 (Bloomfield and Brayshaw, 2021a).

To the authors' knowledge, this is the first time a comprehensive set of high quality *hourly* time series relating to future climate projections has been published, which is specifically designed to support the energy sector. The purpose of this paper is to detail the methods required for processing the climate model data and illustrate the importance of accounting for climate variability and climate change within energy system modelling from sub-national to European scale. While this study is therefore not intended to be an exhaustive analysis of climate impacts, it is hoped that publishing this data will promote greater use of climate data within energy system modelling.



# 1 Introduction

Energy systems are rapidly decarbonising to meet climate mitigation targets such as the Paris Agreement. There are many possible steps to this decarbonising, including: electrifying heat (Eggimann et al., 2020; Kozarcanin et al., 2020) and transport (McCollum et al., 2014; Boßmann and Staffell, 2015; Bellocchi et al., 2020) and installing more renewable energy generation to 
30 provide clean electricity (Zeyringer et al., 2018; Babatunde et al., 2019). There has been a marked uptake of renewable energy generation in the last few years (Abdelilah et al., 2020). These changes in power system composition result in an increasing dependence on meteorological conditions (Bloomfield et al., 2016). The availability of high quality meteorological data for use in energy system modelling has therefore become extremely important.

The relationship between temperature and electricity demand is well documented in the energy-meteorology literature, 
with low temperatures leading to increased demand for heating, and high temperatures resulting in increased demand for cooling (Taylor and Buizza, 2003; Bessec and Fouquau, 2008; Cassarino et al., 2018). Other weather conditions can also impact demand, such as wind speed (via wind chill), relative humidity (via cooling requirements) and incoming solar radiation (via lighting demand; Bunn and Farmer 1985). It is common for this relationship to be modelled using a regression based framework, including exogenous variables such as day of the week and national holidays (Bloomfield et al., 2016; Deakin et al.,
2021). The meteorological relationship between weather and renewable generation is well known, with wind power relating to wind speeds at wind turbine hub-height and solar power generation predominantly relating to the amount of incoming solar radiation on the solar panel. The efficiency of a solar panel is also influenced by the panel temperature, with panels being less efficient at high temperatures (Evans and Florschuetz, 1977). These relationships are however complicated in reality by factors such as grid constraints, unplanned outages, and curtailment of renewable generators.

There are a number of pre-processed meteorological datasets available, providing national level time series of demand, wind power and solar power generation across multiple European countries, though almost all of these focus exclusively on historic conditions. Examples include those available from the Copernicus Climate Change service (CDS, 2021), Renewables Ninja (Staffell and Pfenninger, 2016), and the University of Reading Data Repositories (e.g. Gonzalez et al. (2020); Bloomfield et al. (2020a)). These datasets are also mostly limited to national-scale reconstructions, which previous work has shown can lead to
sub-optimal investment decisions for wind and solar generation within capacity expansion modelling (Frysztacki et al., 2021).

Although the importance of weather to modern day energy systems is well understood (Troccoli et al., 2014), the procedure for converting gridded meteorological data (often output from numerical weather prediction simulations) into time series of energy variables requires specialist climate data knowledge (Bloomfield et al., 2021b). This is partly due to the large meteorological data volumes, which can be stored in potentially unfamiliar file formats (e.g. .grib, .pp or .netcdf). However, a more 
fundamental and scientifically important challenge is the interpretation of climate model simulations. For example, all climate models contain biases and deficiencies that may require calibration and quantification before realistic impact-assessment can be performed. As well as this, multiple types of uncertainty exist within climate modelling. These include: the model setup (such as choice of greenhouse gas emissions pathway, or whether the atmospheric model is coupled to an ocean model) the models representation of internal climate variability, and the model uncertainty (i.e. how the model responds to the changing



radiative forcing). The relative importance of these factors depends on the variables considered, the spatial and temporal scales considered and the lead time of the projection required (Hawkins and Sutton, 2009).

While creating the following datasets, discussions in the energy-meteorology community (further outlined in Bloomfield et al. 2021b) lead to a list of requirements being created. The datasets must:

- **Have hourly temporal resolution:** Power system planning models tend to run at hourly (or sub-hourly) resolution to thoroughly consider potential operational constraints (Collins et al., 2018). For example simulations take into account: technologies for bulk energy storage, changes in diurnal heat demand profiles or sub-daily electric vehicle charging patterns.

- **Provided over a multi-decadal historical period:** Using a *typical-meteorological year* is not enough to account for the impacts of inter-annual climate variability in weather-dependent power systems (Bloomfield et al., 2016; Collins et al., 2018) or to understand the impact of extreme weather events on power system operation (Dawkins et al., 2020; Bloomfield et al., 2020b).

- **Represent the potential impacts of climate change:** It is now a requirement that European Resource Adequacy Assessments to now include the impacts of climate change (ENTSO-E, 2020). The provision of accurate future climate information is now critical for multiple industries including energy.

To the best of our knowledge, there exists no open-source dataset that meets these requirements. Although existing datasets were commonly hourly and multi-decadal, they often did not include the possible impacts of climate change. Alternatively they lacked finer spatial resolution than national level. Having sub-national data allows for a more accurate representation of within-country flows, sensitivity analysis on future renewable generation locations, and other useful information for cost optimisation modelling (Frysztacki et al., 2021). We note the useful work of Bartok et al. (2019) which processed meteorological variables from EURO-CORDEX (Jacob et al., 2014) simulations for use in energy modelling. However, this dataset still requires substantial storage space, and further processing by an end-user to get to regional time series.

When developing these datasets, compatibility issues began to emerge between the available meteorological data, and the requirements listed above. Firstly, although hourly present-day meteorological information is commonly available from meteorological reanalysis products (see section 2.1 for a description of these), hourly future climate simulations are not commonly available. This is partly due to the significant amount of storage space required when running global climate model simulations. However, this also reflects historical practise in climate modelling, where there is scepticism about the usefulness of outputting hourly data given the difficulty of representing meteorological processes at fine spatio-temporal scales. A secondary issue is that energy system modellers may be required to work on small countries (e.g Netherlands and Belgium) or at sub-national level. An example scale here is the nomenclature of territorial units for statistics (NUTS) regions, which can be close to, or smaller than the spatial resolution of climate model simulations (e.g. less than 100km). The amount of information contained in relatively low resolution climate model simulations (such as those from the 6th coupled model inter-comparison project (CMIP6) archive, with 100-250km spatial resolution) could therefore be problematic.



The workflow and resulting datasets presented in this study demonstrate how to create energy-meteorology datasets which can be used to model the impact of present-day climate variability and near-term climate change on European energy systems. The outlined methods are readily extendable to include other regions or climate models using the code provided in the data repositories.

## 2 Data

### 2.1 The ERA5 Reanalysis

The historical meteorological data used in this study is from the ERA5 reanalysis (Hersbach et al., 2020), which is available to download from the Climate Data Store, (CDS, 2021). In essence, a reanalysis is a gridded 3D reconstruction of the past state of the atmosphere. It is created by running a numerical weather prediction model, with data assimilation, to an extensive set of quality controlled observations for the required period. The ERA5 reanalysis is currently available from 1950-present in hourly time steps at 0.3° spatial resolution (around 30km over Europe). The variables taken from the CDS for use in this study are: hourly 2m temperature, hourly accumulations of surface solar irradiance, and hourly 10m and 100m wind speeds. The wind speeds are calculated from the zonal and meridional wind vectors at hourly frequency prior to any subsequent pre-processing at each height.

It has previously been highlighted that ERA5's near-surface wind speeds are subject to some biases compared to other similar products, particularly over mountainous regions (Bloomfield et al., 2020b; Jourdier, 2020). To accurately represent the 100m wind speeds a mean bias correction procedure was applied to adjust the magnitude of the ERA5 wind speeds to those from the Global Wind Atlas dataset (GWA, 2018), as in Bloomfield et al. (2020b). This correction is important for the accurate representation of wind power capacity factors (see section 3.3).

### 2.2 Climate model simulations

A key challenge when selecting appropriate climate models was to find models which output *hourly* 10m wind speeds, surface shortwave irradiance and 2m temperatures. The source of climate data is the PRIMAVERA projects data archive (https://www.primavera-h2020.eu/) which produced a set of high spatial and temporal resolution global climate model outputs. However, of the seventeen model simulations available through the Centre for Environmental Data Analysis (CEDA) archive, (available at https://www.ceda.ac.uk/services/ceda-archive/), only five simulations were available that output all three meteorological variables required to model demand, wind and solar power capacity factors (and even then three of these simulations were at 3-hourly resolution). This highlights a problem in the climate modelling community that the appropriate surface model outputs are still not available for high resolution impact studies. Details of the five available model simulations used are given in Table 1.

The climate model simulations were downloaded over a European domain from: 15W - 40E and 30N - 75N as shown in Figure 1. Table 1 shows that even the highest resolution global climate model simulations are approximately 50km resolution



**Table 1.** Details of climate models used in this dataset

| Model Name | Modeling Centre | Spatial Resolution (km) | Temporal resolution (h) |
|---|---|---|---|
| ERA5 | ECMWF | 30 | 1 |
| MOHC-MM-1hr (ens 2) | UK Met Office | 100 | 1 |
| MOHC-MM-1hr (ens 3) | UK Met Office | 100 | 1 |
| MOHC-HH-3hr | UK Met Office | 50 | 3 |
| EC-EARTH3P-HR | European Community | 50 | 3 |
| EC-EARTH3P | European Community | 100 | 3 |

over Europe, which is coarser than the 30km resolution of ERA5. All of the five model simulations were therefore interpolated
onto the ERA5 grid before calibration and analysis to provide the fairest comparison between datasets.

A number of 3 hourly *regional* climate model outputs are available for analysis using the EURO-CORDEX data (Jacob
et al. 2014 available at https://www.euro-cordex.net/). However, in this case we wished to use global climate model data, to
improve the reproducibility for those wishing to implement the code in other regions (albeit with the limitations of some global
climate models spatial-temporal resolution). It is well known that global scale processes can impact local climate variability
and extremes (see Lledó et al. 2018 and Lledó and Doblas-Reyes 2020 for energy-meteorology relevant examples) but these
are not represented in regional climate model simulations due to the smaller domain size.

Some three hourly global climate model data is available from the sixth Coupled Model Intercomparison Project (CMIP6)
data archive. In using global climate model outputs from the high-resolution PRIMAVERA simulations we have the opportunity
to link to analysis using CMIP6 outputs (even if they are themselves lower resolution in time and space) in the future, to gain
a wider perspective.

### 2.3  Region Masks

Figure 1 shows a schematic diagram of the various zones provided in the ERA5 dataset. Gridded ERA5 data is aggregated
over these regions to resultant time series that can be used as an input for energy system modelling simulations at national or
continental scale. For the onshore regions NUTS0 level data is provided for 38 European countries (see Appendix 1 for a list
of these). The only deviation from these zones is over the United Kingdom, which is instead split into Great Britain (England,
Scotland, and Wales) and then All-Ireland, to better align with the local electricity grid structure. As the GB system was a key
focus in this project, this region is further divided into NUTS1 level and Scotland is provided at NUTS2 level (see Figure 1).
This allows for a demonstration of the value of sub-national data inputs.

To represent all offshore locations where it is possible for a nation to build wind turbines, time series are created for the
Exclusive Economic zones (EEZs) of 25 European countries (Institute 2019, see Figure 1). Finer discretisation was required
around the United Kingdom and the North Sea. The EEZs from the United Kingdom, Ireland and Norway were therefore sub-

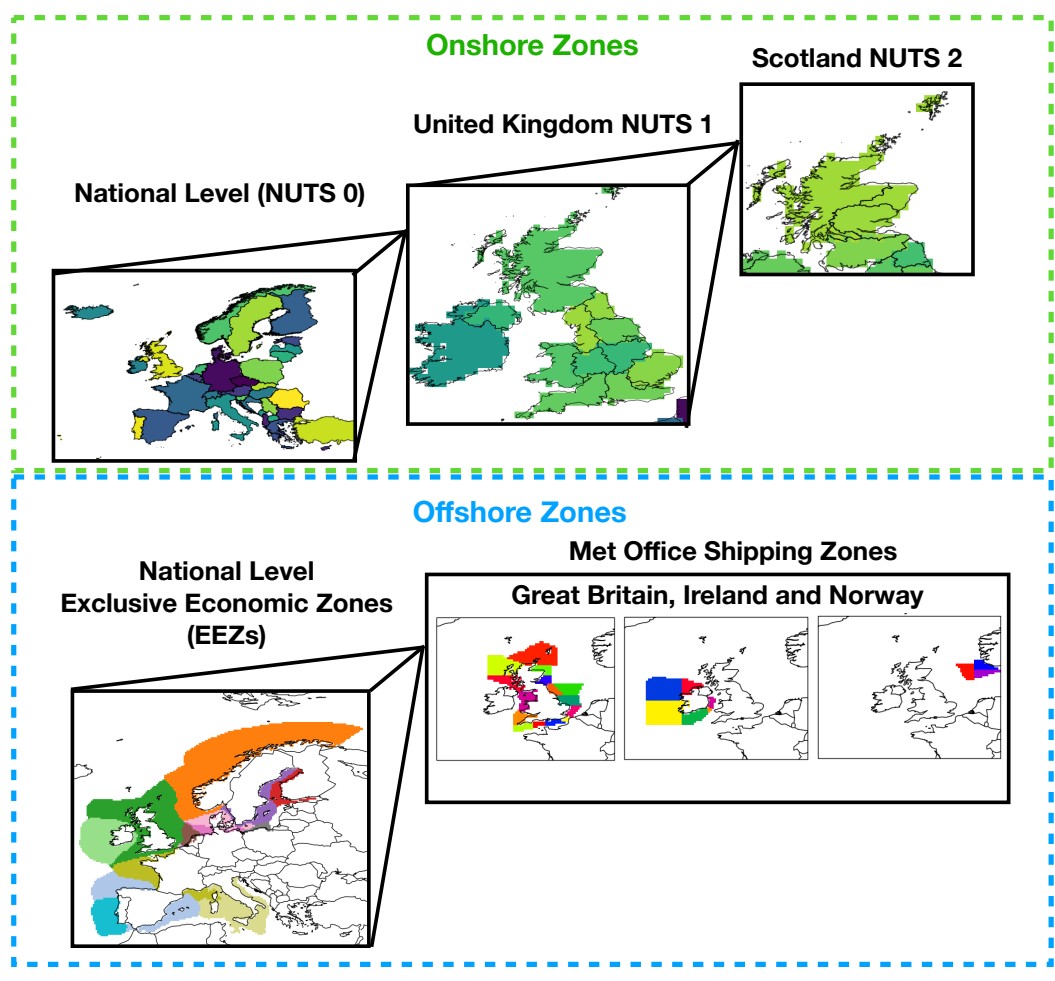

**Figure 1.** A schematic of the onshore (green box) and offshore (blue box) zones used in this study to aggregate meteorological variables and capacity factors over.

set by the UK Met Office Shipping forecasting zones, which align well with the placement of many current and planned large offshore wind farms. Although the UK has been used here as a demonstration of high-resolution offshore generation zones, these are available worldwide and could be implemented for future studies. Appendix 1 gives further details of these zones.

Within each zone described above the appropriate meteorological data, or capacity factors are given a set of weights to aggregate to a time series. Equal area weightings are provided for all regions for hypothetical studies which are not focused

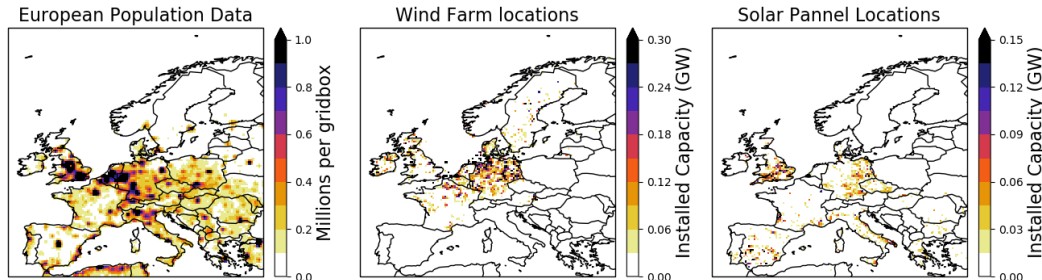

**Figure 2.** (a) Population data (b) wind farm locations (c) solar farm locations over Europe used to create the datasets in this study, all interpolated onto the ERA5 reanalysis grid. See Appendix 1 for further details of the fields produced in each zone.

around existing or future renewable generation locations. Population-weighting is provided to the meteorological fields as this is useful for modelling energy demand (for example in large countries like Norway, the country-average temperature is not representative of the relationship between temperature and demand as very few people live in the Northern latitudes). Population data is taken from Doxsey-Whitfield et al. (2015). Weightings based on existing wind and solar farms are also included. Installed wind power generation is taken from thewindpower.net database. Solar panel locations are taken from Dunnett et al. (2020), except for GB where a recent dataset from Stowell et al. (2020) is used, as this improves the solar model performance (not shown). Solar panel locations and population data are kept at 2021 levels for all of the datasets that are produced.

The non-meteorological datasets used in this study for the 2021 period are shown in Figure 2. These have all been interpolated on the grid of the ERA5 data used in this study. TheWindPower.net database also includes some information on the location of proposed future wind farms and those currently under construction (not shown). A future wind power simulation is also included for the wind power capacity factor datasets for countries where this data is available.

## 3 Methods

### 3.1 Calibration of climate model data

When investigating the impacts of climate change on any impact-system it is important that the underlying climate model data does not give a biased representation of the present day climate, as this may result in an incorrect interpretation of the results. One of the simplest and most common methods of climate model calibration is known as delta correction (Hawkins et al., 2013). The delta correction method has a long history in climate impact research (Belcher et al., 2005; Maraun, 2016). Rather than performing a bias correction on the climate model data itself, this method applies a modelled climate change response to adjust a set of pre-existing historical observations. One key merit of the delta correction is that it is immediately interpretable by a non-expert audience, due to it providing adjustments to known past weather events. This is opposed to a bias adjustment





technique which scales the climate model data – which does not relate to past weather events - to remove biases present in the historical period compared to observations (see Bartok et al. 2019 for a demonstration of this). A key area of interest within

energy meteorology is the impact of extreme weather events on potential power system operation (Bloomfield et al., 2021b). A delta correction can therefore show potential impacts of climate change on past extreme events (see section 4.2).

A description of an additive delta correction in its simplest form is shown in Equation 1:

$$OBS_{delta}(t) = OBS_{hist}(t) + (\overline{MOD}_{fut} - \overline{MOD}_{hist}) \tag{1}$$

Here t is the hourly time step. The historical (hist) and future (fut) periods are an average over multiple decades to get a mean

response and OBS is the observations. A different correction can be applied for different seasons, as the projected change can vary over each meteorological year. Adaptations can also be included to correct both the mean and variance of a dataset (see Maraun 2016). Quantile mapping corrects both the mean and variance by applying a different magnitude of correction at different percentiles (Maraun, 2016). Preliminary analysis of the five climate models from Table 1 found they are able to represent the present-day seasonal cycles of the near-surface weather variables well, with some small biases. The diurnal cycles

of near-surface weather variables were also reproduced well by the models (not shown) lending some confidence to the choice of the delta correction approach. An example of the gridded seasonal-mean delta correction factor between 1980-2010 and 2020-2050 periods are shown in Figure 3 for the five climate model simulations. This period was chosen for its relevance to power system infrastructure planning, and as a period where the large uncertainty in energy system design does not dwarf the possible impacts of climate change (see Bloomfield et al. 2021a for examples).

Detailed analysis found the impact of climate change on the extremes of the meteorological variables were often different to the mean response within each season. For this reason the seasonal delta correction factors were applied to each percentile of the distribution. Examples of this are shown in Figure 4 for one climate model, at the grid points of ERA5 closest to three large European cities. For this chosen model, London summer temperatures experience larger impacts of climate change than winter temperatures. Whereas, in Helsinki winter temperatures (particularly in the lower percentiles of the distribution) require

larger corrections than the summer ones.

A seasonal quantile based correction has been implemented in this study to allow a detailed focus on extreme events which may impact the energy sector throughout the year. In this study delta corrections are applied to the gridded hourly ERA5 data to represent the impacts of climate change. Our particular delta correction takes the form shown in Equation 2:

$$ERA5_{delta}(x,y,season,t) = OBS_{hist}(x,y,season,t) + (\overline{MOD}_{fut}(x,y,season) - \overline{MOD}_{hist}(x,y,season)) \tag{2}$$

Where x and y are the latitude and longitude of each grid point the correction is performed on respectively and other variables are as described in Equation 1. The correction is applied before computing area-aggregated variables. The delta corrections for 10m wind speed are used on the 100m wind speed field for ERA5, as this is the closest height available in the climate model output.

The correction here presents a different approach to a similar problem that was tackled in Bartok et al. (2019). This is an

area where substantially more research is needed to work out the merits and issues of various correction methods for use in power system modelling.



## 3.2 De-trended temperature data

As the historical period of ERA5 spans from 1950-2020 some statistically significant impacts of climate change are already apparent over European land, particularly for 2m temperatures. Due to this a de-trended version of the 2m temperatures from
210 ERA5 is also provided where a linear model is fitted from 1950-2020 to remove the long-term trend. The data is then scaled to be representative of background temperature field from: 1950, 1980 and 2010. This is useful for studies wanting to include the impacts of year-year climate variability but that may have concerns about the plausibility of past events with current levels of warming.

## 3.3 Wind Power Capacity Factor

A physical model is used to produce estimates of regional wind power capacity factor. Gridded 100m wind speeds from the ERA5 reanalysis are converted into wind power capacity factors using either an onshore or offshore power curve extracted from a National Grid report (National Grid), and shown in Figure 5. Sensitivity testing was conducted, comparing to curves used in Bloomfield et al. (2020b) to confirm these gave an improved fit to measured wind power generation from ENTSOE (2020) across multiple European countries compared to results from Bloomfield et al. (2020b).

Before passing the gridded 100m wind speeds through the appropriate power curve they are scaled to the UK average onshore or offshore hub-height using equation 2.

$$U_{hub} = U_{100m}(\frac{Z_{hub}}{Z_{100m}})^\alpha \tag{3}$$

Here U is the wind speed, Z is the height from the surface and $\alpha = \frac{1}{7}$, which is an empirically derived coefficient related to the stability of the atmosphere. This method is commonly used in the wind power modelling community (e.g. Lledó et al. (2019);
Bloomfield et al. (2020b)). The onshore and offshore hub-heights were 71m and 92m respectively. These were calculated as the weighted average of operational wind farms in April 2021.

Information regarding the spatial distribution, hub-heights and installed capacity of wind turbines is taken from $thewindpower.net$ database, with the current wind farm fleet representing those producing electricity in April 2021 (see Figure 2). The models perform well compared to others in the literature, with an average daily $R^2$ of 0.95, and average percentage error of $12\%$ when
validated against data from ENTSOE (2020). Validating a physical wind power model against measured data is complex due to the models inability to represent grid constraint, maintenance periods or wind power curtailment. For this reason day ahead forecasts from the ELEXON portal (Elexon, 2021) are used for model verification in GB, an example of these are shown in Appendix Figure A3.

## 3.4 Solar Power Capacity Factor

The solar PV model follows the empirical formulation of Evans and Florschuetz (1977) but with adaptation to newer solar PV technologies using methods from Bett and Thornton (2016) and Bloomfield et al. (2020b). The meteorological inputs are



gridded 2m temperature (T) and incoming surface solar irradiance (G), from which hourly solar power capacity factor (CF) is calculated using the equation below:

$$CF(t) = \frac{power}{power_{STC}} = \eta(G,T)\frac{G(t)}{G_{STC}(t)} \tag{4}$$

Here G is the incoming surface shortwave radiation and T is 2m temperature, and t is the time step (hours). STC stands for standard test conditions (T = 25°C G = 1000 Wm$^{-2}$) and $\eta$ is the relative efficiency of the panel following

$$\eta(G,T) = \eta_r[1 - \beta_r(T_c - T_r)] \tag{5}$$

Where $\eta_r$ is the photovoltaic cell efficiency evaluated at the reference temperature $T_r$, $\beta_r$ is the fractional decrease of cell efficiency per unit temperature increase and $T_c$ is the cell temperature (assumed to be identical to the grid box temperature).
The model performs well with an average daily $R^2$ of 0.93 with 4.8% error for countries where data was available at high enough quality for validation from the ENTSOe transparency portal ENTSOE (2020). Further details of the model formulation and verification can be found in Bloomfield et al. (2020b). An example of the model performance for GB is given in Figure A3.

## 4 Results

In this section some illustrative examples of the two datasets are given. Firstly the benefits of some of the sub-national data outputs are demonstrated in section 4.1. Following this the possible impacts of climate change on notable past events and compound impacts are discussed in section 4.2. Finally the impact of climate model selection is briefly discussed (section 4.3).

### 4.1 What can we learn with sub-national data outputs?

One of the key advancements of this dataset compared to many of its predecessors is the inclusion of sub-national time series,
which can be particularly useful for countries spanning a large geographical area with significant renewable resource diversity. For example, GB has plans to significantly increase offshore wind capacity, with a very different spatial distribution to today's fleet (based on proposed farms from thewindpower.net database). Offshore floating designs also have potential to exploit areas that were previously infeasible (Moore et al., 2018).
  Figure 6 shows an example of how the GB nationally-aggregated offshore wind power capacity factor (weighted by the
260 location of current wind farms shown in Figure 2) compares to the capacity factor averaged over three of the UK Met Office shipping zones. The shipping zones are chosen for the spatial diversity around GB and for the large amount of wind power generation installed in each one (see caption of Figure 6). The period chosen is a cold-snap through December 2020 where generation from wind power was needed to meet anomalously high demand. Although the modeled national capacity factor is quite high throughout the period, we see that there are marked drops on the capacity factor in the Irish Sea on Christmas





265  day, and in Cromarty on the 28th December. Considerably more ramping is seen at this finer spatial scale, which could create challenges for grid balancing across GB.

Figure 7 shows the Pearson correlation coefficient between the GB total offshore wind generation and the capacity factor from the UK shipping zones throughout the whole reanalysis period (1950-2020). Neighbouring zones wind power capacity factor have highest correlations. When comparing to Figure 6 we can note that the low correlation between Thames and 270  Cromarty (0.29), but higher correlation between Thames and the Irish Sea (0.49) and Cromarty and the Irish Sea (0.59).

Similar analysis can be completed for solar power capacity factor across UK NUTS1 and NUTS2 zones (similar results are seem for 2m temperature and degree days; not shown). Figure 7 shows higher correlations for the UK solar capacity factors than seen for wind generation due to the more pronounced diurnal and seasonal cycles. Again it is notable that neighbouring regions have highest correlation, and the lowest correlations are seen between Scotland (the UKM zones) and Southern England (UKI 275  and UKJ zones). Analysis of this type can be useful to understand potential strains on existing grid infrastructure and can be used to think about optimal future wind and solar farm distributions.

### 4.2   How could climate change impact past power system extremes?

One potential use of the delta-corrected climate data it to revisit notable past weather events and see how they may be influenced by climate change. Figure 8 shows two examples of extreme events. The February 1963 big freeze was previously highlighted 280  as being one of the coldest winters in the last 80 years by Cattiaux et al. (2010), who used it as a benchmark for the more recent 2010 cold event. The impact of a winter with extreme prolonged low temperatures could cause challenges for maintaining security of supply. Within the ERA5 data we see the GB population weighted temperature stayed below 2 degrees for over a week (black line in Figure 8), resulting in persistent snowy conditions. If this event were to happen with the background warming level expected by 2035 (coloured lines in Figure 8) we see this event would still have been exceptionally cold, but 285  the minimum temperatures reached would not have been as low as seen in 1963, implying a slightly reduced strain on the electricity grids. An important caveat here is that by 2035 we would expect that the proportion of electric heating to have increased, resulting in a stronger relationship between temperature and demand. This change in system composition could mean that even in a warmer climate the winter of 1963 could still be a challenging peak demand event (Deakin et al., 2021).

The second notable event is the 2003 heatwave. This was a period of prolonged high temperatures over Europe, Temperatures 290  were the highest seen since the start of the instrumental record in 1851 (Stott et al., 2004) which if occurring in a present day energy system could result in increased demand for air conditioning and increased cooling water requirements in traditional thermal power plants. Figure 8 shows that the GB population weighted temperatures at the peak of the diurnal cycle are exceeding 30 degrees if the impact of potential near-future climate change is included (coloured lines). In this event it is clear to see the sensitivity to the quantile based delta corrections, as the high temperatures experience a much larger correction 295  than the more moderate ones. An area worthy of future investigation to understand how large the differences between climate change projections are compared to the uncertainty that arrives from the choice of delta-correction procedure (e.g. a percentile based vs. mean delta correction).





Using the 70-year delta corrected dataset it is also possible to look at the change in frequency of extreme events. Here we have chosen to focus on hot days, which are defined as the exceedance of the 90th percentile of ERA5's daily-mean 2m

temperature. Figure 9 shows how many extra summertime (June-August) hot days are seen in the delta corrected ERA5 data for each of the five model simulations. It is notable that the UK Met Office simulations (MOHC runs) have a larger summer warming than the EC-EARTH model runs. This was also evident in the previous case studies (Figure 8) and Figure A1. The response of 10 extra hot days (when averaged over all 5 model simulations) is quite uniform across the Central-Northern European countries analysed.

Extra hot days could result increased demand. However, Figure 9 shows that of these extra hot days only around 15% of these are both hot and still (2m temperature exceeding the 90th percentile in ERA5 and 10m wind speeds below the 10th percentile) suggesting that there may be wind power generation available to meet elevated demands in general. Only 5% of the days are hot and cloudy (2m temperature exceeding the 90th percentile in ERA5 and surface solar irradiance below the 10th percentile) suggesting a good complementarity between solar power and periods of elevated demand. We note there that

these results are a set of first impressions of the dataset and require further scientific analysis to fully understand the potential impacts on society.

## 4.3 The impact of climate model selection

In this study five climate model simulations have been used from two modelling centres (see Table 1 for full details). It is beyond the scope of the present paper to provide a comprehensive discussion of the differences between the climate model

projections. However, the importance of using both multi-model and multi-realisation climate model archives for power system applications are illustrated by the following observations.

When initially analysing Figure 3 A1 and A2, it appears that all five simulations have a broadly similar response to climate change. All models 2m temperatures increase in a future climate, with the largest increase seen in summer (Figure A1) Similar results are seen for surface solar irradiance over central Europe (Figure 3). However, there are some quantitative differences

in the projected responses. The two different modelling centres (EC-EARTH vs. MOHC) produce opposite winter wind speed projections over the North Sea, with increases seen in the Met office Model and decreases seen in EC-EARTH (Figure A2). The differences between models is also seen in Figure 9 where the UK Met office model runs (orange and red colours) experience double the amount of hot days compared to the EC-EARTH model across Europe.

The second observation is that multiple runs with the same model setup (e.g. the two MOHC-MM-1hr simulations) can give

different results. An example of this is seen in Figure A2 where spring average wind speeds show opposite signs of change over the North Sea and central Europe. Given that these changes are both statistically significant, this suggests that the differences are likely due to internal variability (see Bloomfield et al. (2021a) for further discussion) and require careful interpretation. With these short (30-year) samples used to create the climate change impact, it is difficult to attribute any differences between simulations to changes in model resolution, but this ongoing analysis in the meteorological community (see Bador et al. 2020).

This shows the importance of considering multiple possible future simulations and future work could extend this subset of models even further when more high resolution data is available through future projects.



# 5    Conclusions

This paper showcases a new dataset of European meteorological variables and weather-dependent energy variables for use in energy systems modelling, providing information on both past (1950-2020) and future (2020-2050) climate conditions. The
dataset provides time series over a number of national, regional, and offshore zones.

The future climate data corresponds to a set of climate-change-corrected 70 yer climate samples (based on 5 different climate model projections) each representing the climate epoch 2020-2050. This allows the impacts of climate change from five different high resolution climate models to be investigated. The climate change impacts are centred on 2035 to maximise their relevance to near-future infrastructure planning while ensuring that the uncertainty in energy system design does not
dwarf the possible impacts of climate change.

Examples of the datasets are shown in the results section to display the different modelled regions as well as the possible impacts of climate change on past notable events, and compound extremes. Future work could include a much broader spectrum of outputs such as data from the sixth full Coupled Model Intercomparsion Project (CMIP6) archive and EURO-CORDEX to compliment the work of Bartok et al. (2019).

It is hoped that publishing this data will promote the uptake and use of state-of-the-art climate data within energy system modelling. The code to create all of the data outputs from this project has been made available and the authors welcome discussion from users in other impacts fields, or regions of the globe.

# 6    Code and data availability

Hourly, time series of 2m temperature, 10m wind speed, 100m wind speed, surface solar irradiance, degree days, wind
power capacity factor and solar power capacity factor from 1950-2020, derived from the ERA5 reanalysis are available at https://researchdata.reading.ac.uk/321/ with Pre-assigned DOI https://doi.org/10.17864/1947.000321, which can be activated after review (Bloomfield and Brayshaw, 2021b). Future climate projections of surface weather variables, wind power, and solar power capacity factors across North-West Europe and the code used to create time series used in the paper are available from https://researchdata.reading.ac.uk/331/ with pre-assigned DOI https://doi.org/10.17864/1947.000331 that can be activated after
review (Bloomfield and Brayshaw, 2021a).

The Global Wind Atlas data used to calibrate the ERA5 100m wind speeds is available at https://globalwindatlas.info/ download/gis-files.

The exclusive economic zones used in this project are taken from https://www.marineregions.org/downloads.php The shape-files of Met Office shipping zones were provided by the UK Met Office on request.

**Appendix A:  Details of aggregated regions**

In the ERA5 dataset national level 2m temperature, 10m wind speed, 100m wind speed, heating degree days, cooling degree days, wind power capacity factor and solar power capacity factor data are created for the following countries: Austria, Albania,

Belarus, Belgium, Bosnia and Herzegovina, Bulgaria, Croatia, Czech Republic, Denmark, Estonia, Finland, France, Germany, Greece, Hungary, Ireland, Italy, Kosovo, Latvia, Lithuania, Luxembourg, Macedonia, Moldova, Montenegro, Netherlands, Norway, Poland, Portugal, Romania, Serbia, Slovakia, Slovenia, Spain, Sweden, Switzerland, Turkey, Ukraine, and the United Kingdom.

Location weighted onshore and offshore wind power generation is created for: United Kingdom, Ireland, Netherlands, France, Belgium, Germany, Denmark, Norway, Sweden, Austria, Estonia, Lithuania, Latvia. Nomenclature of territorial units for statistics (NUTS) 1 and 2 regions included in this study are: UKC, UKD, UKE, UKF, UKG, UKH, UKI, UKJ, UKK, UKL, UKM, UKN and UKM5, UKM6, UKM7, UKM8 UKM9 respectively.

Offshore wind power generation averaged over exclusive economic zones is calculated for: France, Italy, Portugal, Estonia, Latvia, Lithuania, Croatia, Romania, Slovenia, Greece, Montenegro, Albania, Bulgaria, Spain, Norway, United Kingdom, Ireland, Finland, Sweden, Belgium, Netherlands, Germany, Denmark and Poland.

Within the exclusive economic zones of the United Kingdom offshore wind power generation is calculated for the following Met Office Shipping Zones (see https://www.metoffice.gov.uk/weather/specialist-forecasts/coast-and-sea/shipping-forecast for more details). For the United Kingdom: Forties, Cromarty, Forth, Tyne, Dogger, Fisher, Humber, Thames, Dover, Wight, Portland, Plymouth, Lundy, Irish Sea, Malin, Hebrides, and Fair Isle. For Ireland: Lundy, Fastnet, Irish Sea, Shannon, Rockall and Malin. Finally, for Norway: South Utsire, Forties and Fisher.

For the datasets including delta corrections the above descriptions are limited to data from the following countries: United Kingdom, Austria, Belgium, Denmark, Finland, France, Germany, Ireland, Netherlands, Norway, Sweden, Latvia, Lithuania and Estonia.

*Author contributions.* Bloomfield created the datasets discussed in this study and wrote the first manuscript draft. Brayshaw supervised Bloomfield and advised on access and processing of climate data. Deakin and Greenwood provided expert advice on the requirements for the dataset creation and were involved in the wind/solar model development, quality control and testing of the produced datasets.

*Competing interests.* No competing interests are present

*Acknowledgements.* This work was funded as part of the CLimate-Energy modelling for Assessing Resilience: HEAt Decarbonisation and the Northwest European Supergrid (CLEARHEADS) project.

We thank Paula Gonzalez for advice when using the PRIMAVERA data archive, Humphrey Lean and Neil Armstrong for the assistance in locating Met Office shipping zone shapefiles and to Jon Seddon for providing access and information on the hourly Met Office climate model outputs used in this study.



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

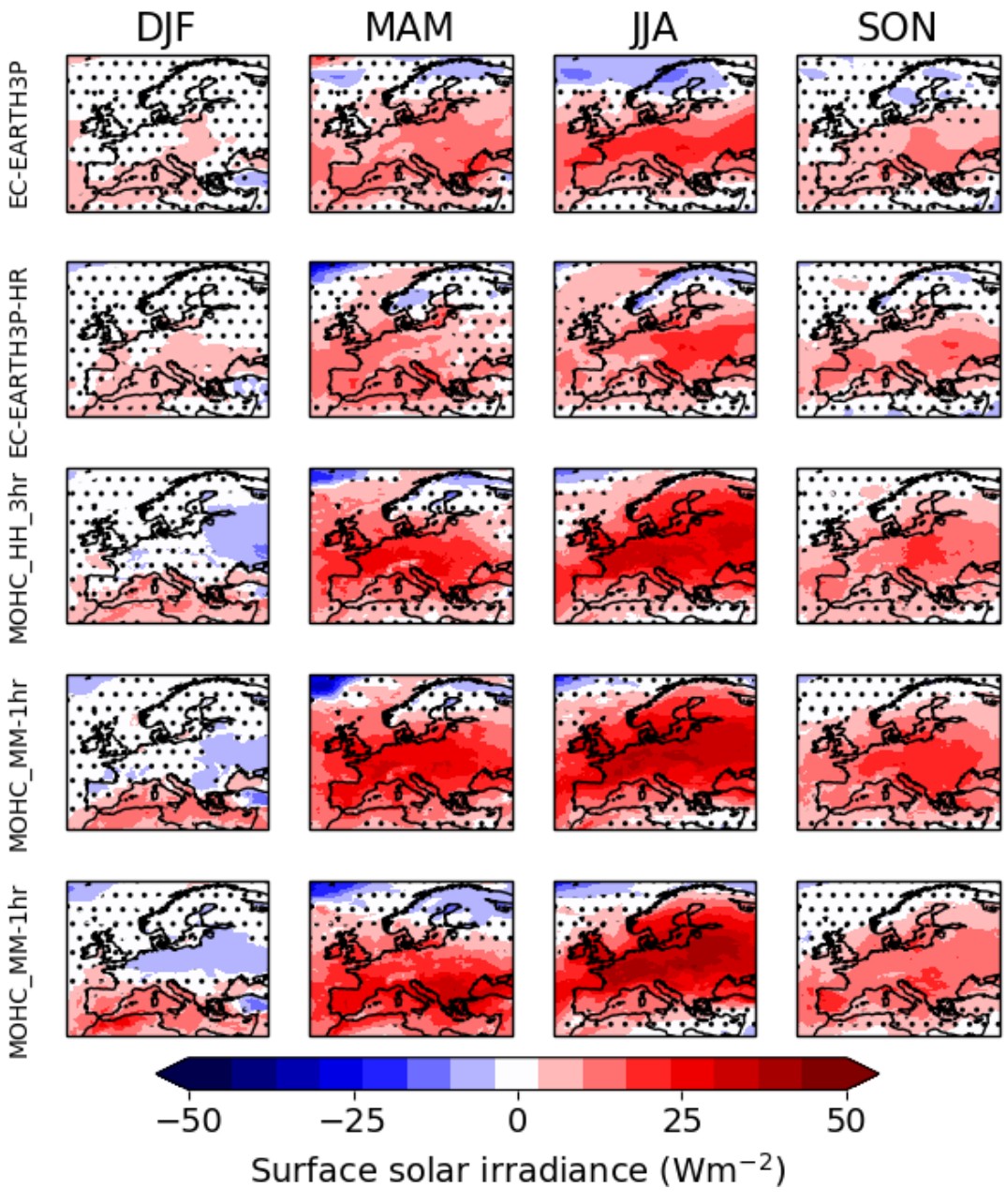

**Figure 3.** Seasonal-mean difference in surface solar irradiance between 1980-2010 and 2020-2050 for the 5 different climate model simulations used in this study (see Table 1 for further details of the models. Stippling shows grid points where no statistically significant climate change signal was found when using a 2-sample t-test (Wilks, 2011).

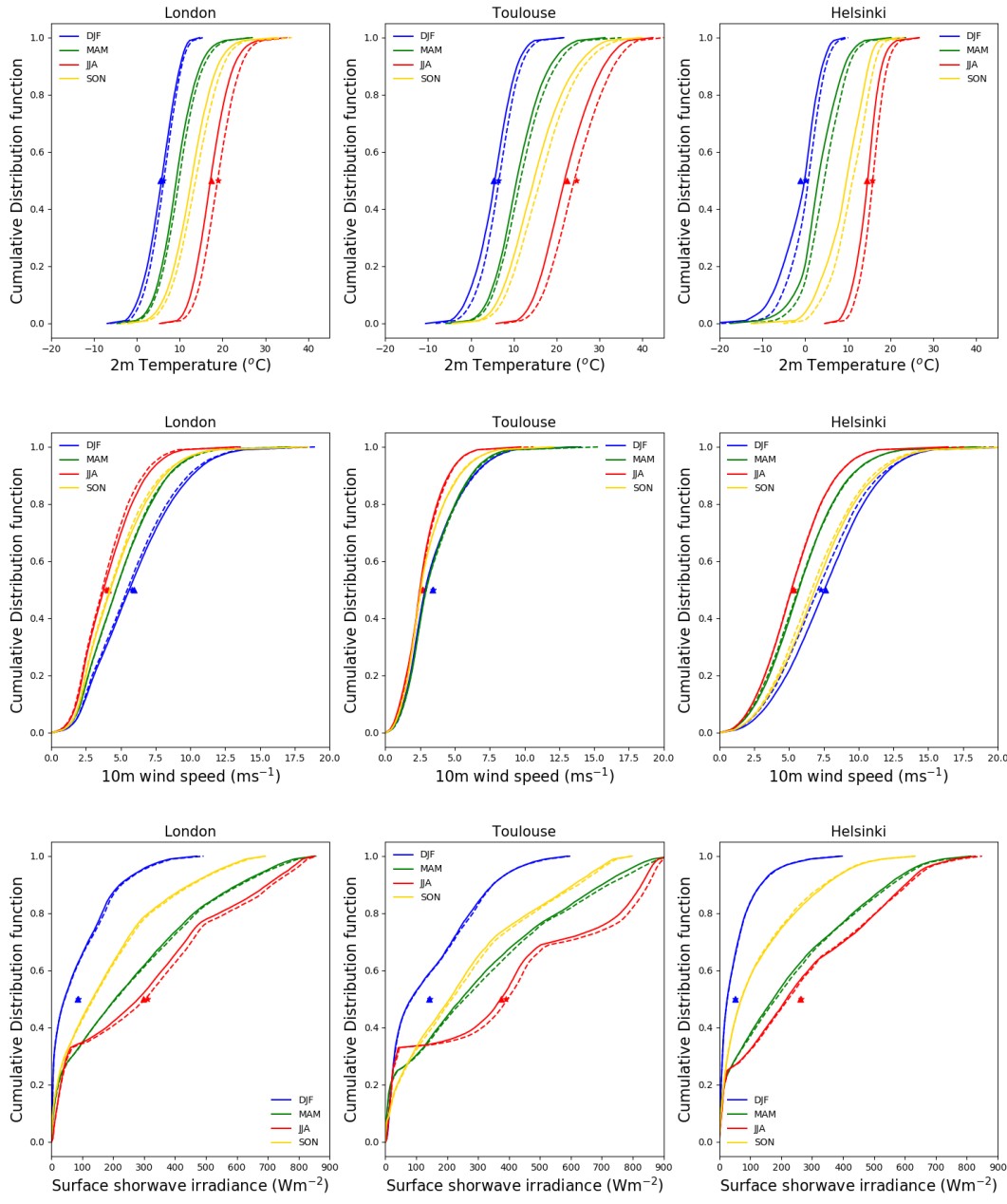

**Figure 4.** Seasonal percentile distributions of (top) 2m temperature (middle) 100m wind speed and (bottom) surface shortwave radiation for the nearest grid point to 3 major European cities. Solid lines show ERA5, dashed lines show ERA5 including the impact of climate change from the ECEARTH-3P model.

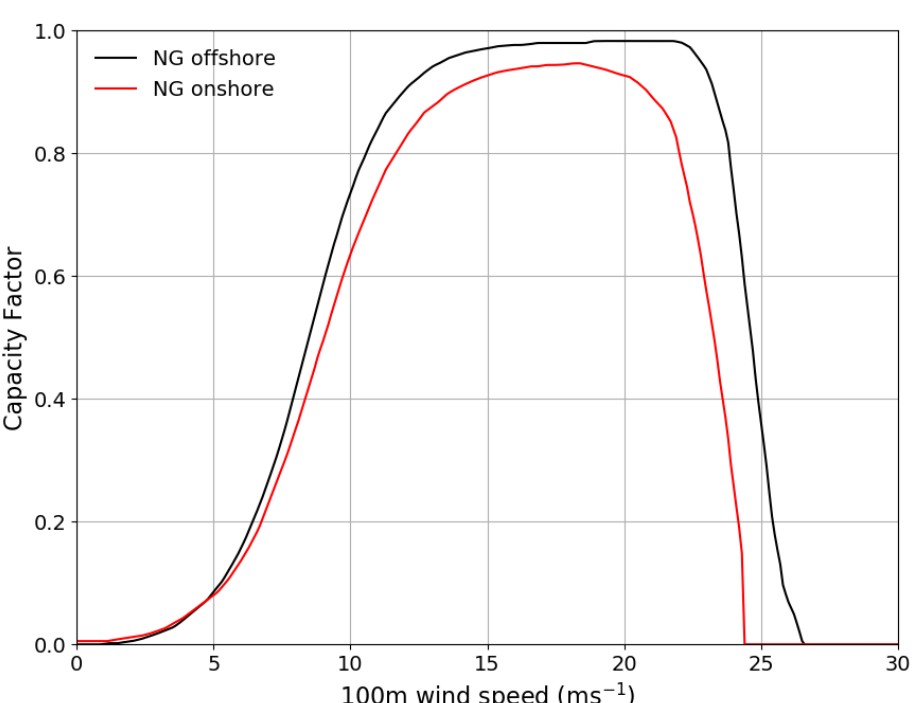

**Figure 5.** The onshore (red) and offshore (black) wind power curves used in this study, taken from National Grid

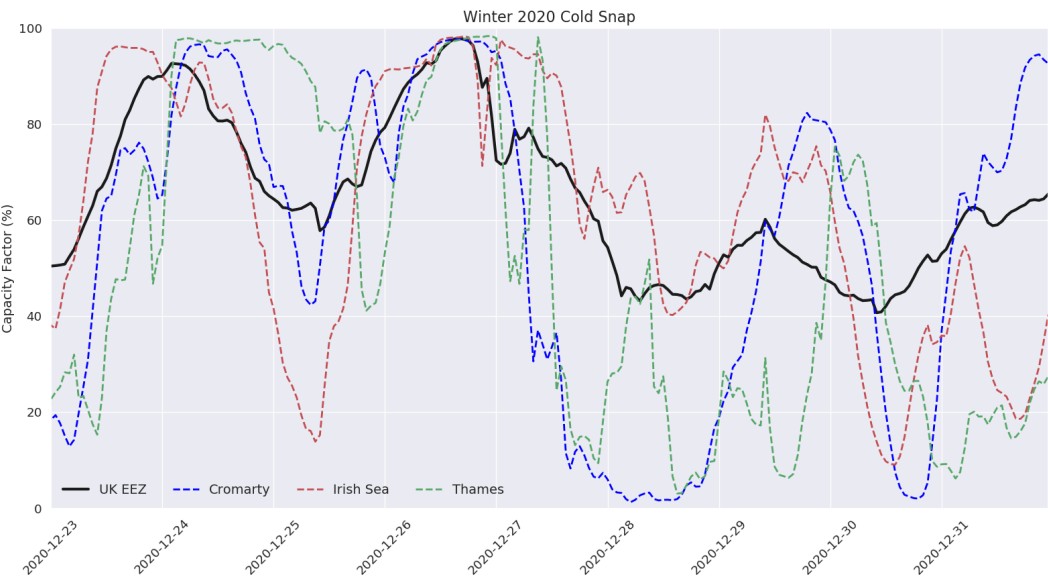

**Figure 6.** Case study of a period of variable capacity factor showing national time series over GB offshore wind power capacity factor (black) and three sub-regions containing a large proportion of UK offshore wind: Cromarty (blue; 0.7GW) Irish Sea (red; 1.7GW) and Thames (Green; 2.3GW)

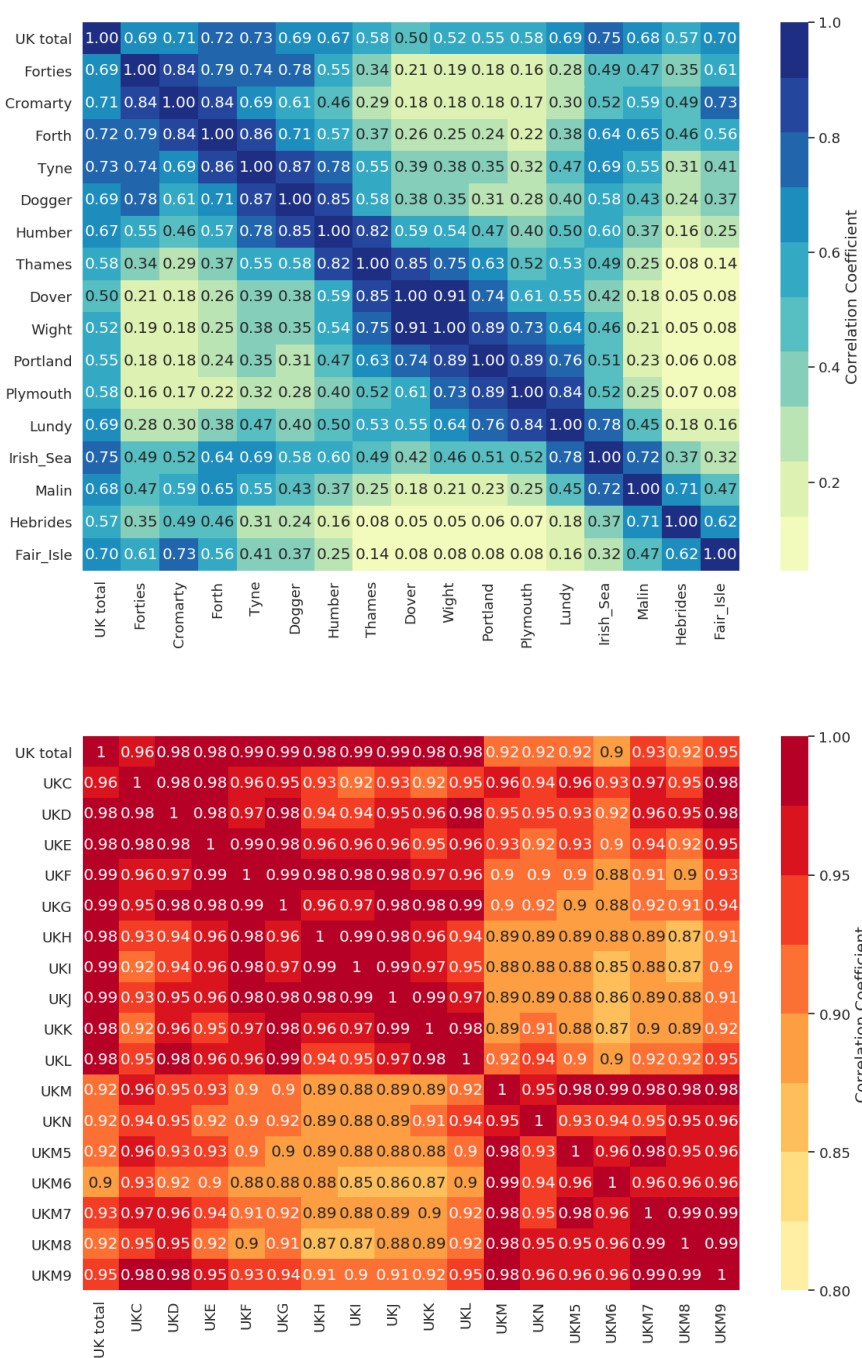

**Figure 7.** Pearson Correlation coefficient heat map of (top) hourly offshore wind capacity factor time series from the Met Office Shipping Zones compared to the UK location-weighted total. (bottom) hourly solar power capacity factor for NUTS1 and NUTS2 zones compared to the UK location-weighted total.

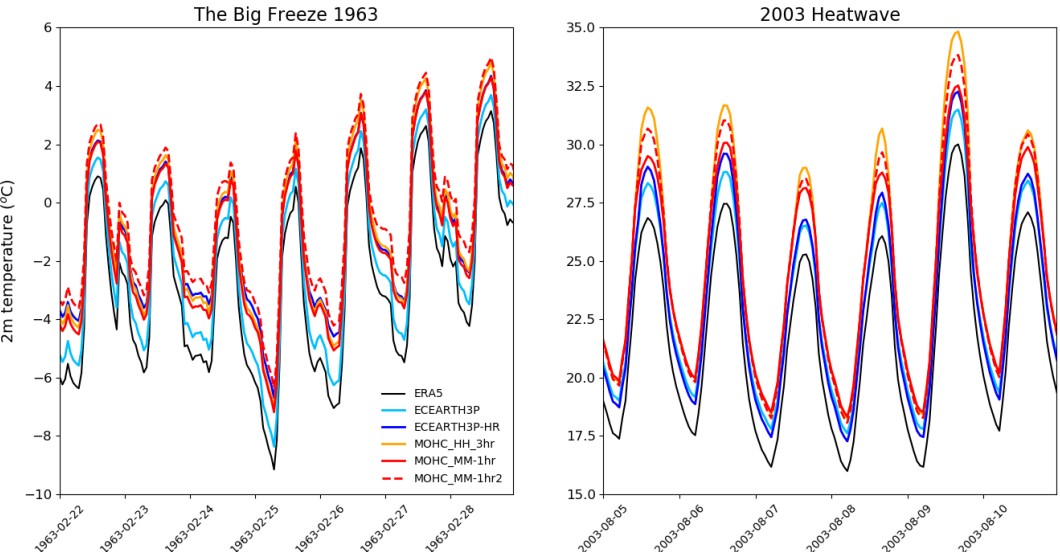

**Figure 8.** Top panels: Two case study events showing the possible impacts of climate change on GB population-weighted 2m temperatures. ERA5 is shown in black with delta corrected model projections in colours. See Table 1 for further details of the models.

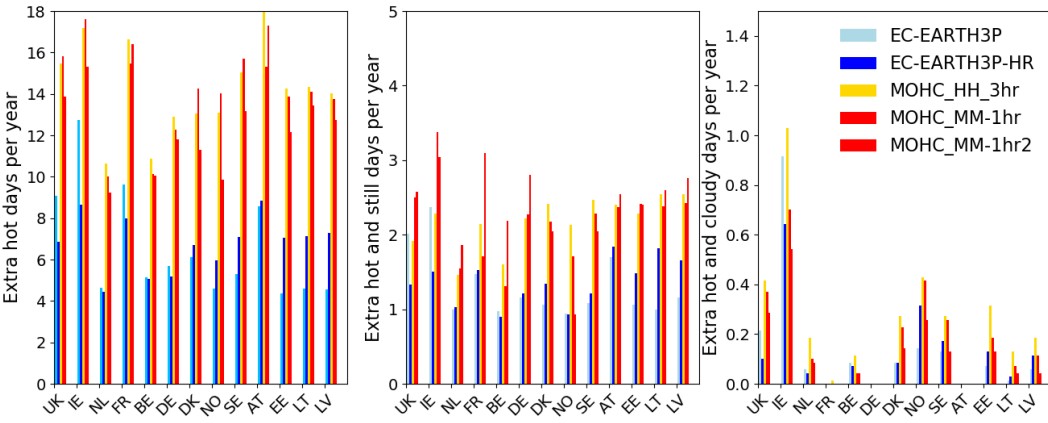

**Figure 9.** (left) The number of extra days per summer (June-August) in a possible future climate where daily Population weighted 2m temperature exceeds the 90th percentile value. (middle) Number of days where 2m temperature exceeds 90th percentile and 10m wind speed is below the 10th percentile. (right) Number of days where 2m temperature exceeds 90th percentile and surface solar irradiance is below the 10th percentile. See Table 1 for further details of the models.



**Figure A1.** Seasonal-mean difference in 2m temperature between 1980-2010 and 2020-2050 for the 5 different climate model simulations used in this study (see Table 1 for further details of the models). Stippling shows grid points where no statistically significant climate change signal was found when using a 2-sample t-test (Wilks, 2011).

Earth System **Science** Data
Open Access · Discussions



**Figure A2.** Seasonal-mean difference in 10m wind speed between 1980-2010 and 2020-2050 for the 5 different climate model simulations used in this study (see Table 1 for further details of the models). Stippling shows grid points where no statistically significant climate change signal was found when using a 2-sample t-test (Wilks, 2011).

**Figure A3.** Example verification of the wind and solar PV models for (top) onshore wind (middle) offshore wind (bottom) solar PV. For the wind power model validation data (black) is taken from the ELEXON data portal, and for the solar PV data it is taken from ENTSOE (ENTSOE, 2020).