# Peer review of "Hourly historical and near-future weather and climate variables for energy system modelling"

_Earth System Science Data, 2021_

## Author Comment (AC1)

**Response to Reviewers** **Hourly historical and near-future weather and climate variables for energy system modelling**

**Reviewer 1**

**General Comments**

The data is accessible from the provided links and in good shape. I was able to download the data, load it into memory and inspect it. It would be helpful to mention that the climate projected data comprises "only" 3 years.

Thank you for the feedback. The future climate projections datasets provide data from 1950-2020 adjusted to represent the future climate of 2020-2050. So when the data is downloaded within the netCDF files you get access to the ERA5 timeseries (form 1950-2020), and then the output from the 5 climate models adjusted using the method from section 3.1., resulting in 5 future climate scenarios. An example of the climate impacts is given in Figure 8. We have checked the dataset descriptions in the paper to make sure this is now clear for future users.

The major drawback is the confinement of the sub-national datasets to the UK (and Ireland for offshore). However the data will be useful for studies focused on UK and potentially the North-Sea area where major investments in offshore wind power are to expect in the upcoming years.

We agree that this is a restriction of the dataset, but one that had to be made due to the limited scope and duration of our funded project. This is however why we have pushed to publish the python code used to run this workflow and the sources of the shapefiles used to create the datasets so others could reproduce the method for future studies across Europe. Assistance can also be provided from the authors in implementing this on request. Following further reviewer comments we have also updated the python code to make sure it is using all currently available libraries for python3.

**Specific comments**

* Introduction: it was not mentioned that their exist automated tools that convert different kind of meteorological datasets into spatially resolved time-series for renewable technologies, like `atlite` (<https://doi.org/10.21105/joss.03294>) or `pvlib` (<https://doi.org/10.21105/joss.00884>). These may potentially allow for processing climate projected data.

Thank you for highlighting these important tools. We've added in reference to these in line 51:

A number of automated tools have now also been developed to convert gridded meteorological data into timeseries of renewable generation (e.g. atlite \cite{hofmann2021} and pvlib \cite{holmgren2018})

* Introduction: There have been other approaches made to use climate projected data from EURO-CORDEX for energy system modelling, e.g. (<https://www.sciencedirect.com/science/article/abs/pii/S0306261918313953>)

Thanks for highlighting this. We have included the along with the Moekmen et al., reference.

We note the useful work of \citet{bartok2019} which processed meteorological variables from EURO-CORDEX \citep{jacob2014} simulations for use in energy modelling. However, this dataset still requires substantial storage space, and further processing by an end-user to get to regional time series. \textcolor{red}{\cite{schlott2018} took meteorological data directly from selected EURO-CORDEX ensemble members, to use as inputs for an energy system model after conversions to energy variables. These future climate projections were used to understand the impacts of climate change on the European energy network. However, climate model inputs were not calibrated before use, which could lead to errors in the results, due potential model errors in the representation of surface meteorological variables. These could manifest in issues with creation of wind and solar PV output due to the strongly non-linear relationships.}

* For now the geographical scope of the national and sub-national dataset are diverging. The sub-national dataset only includes GB (and Ireland for offshore). However, for European energy system models we need a sub-national resolution across multiple countries. For a future project it would be helpful to create datasets on NUTS1/NUTS2 resolution for same set of countries as included in the NUTS0 dataset.

We totally agree that this would be very useful, and hope this work will demonstrate the need for this data to be rolled out further in the future.

* The section 4.2. "How could climate change impact past power system extremes?" is a bit poor. The authors set a strong focus on the extreme temperature periods. However it is rather the interplay between renewable power potential and the demand that is important here. Dark, cold periods with weak wind potential are the most challenging for the (renewable) energy supply. How and to what extent are these changing in the climate projections?

We agree that the winter low renewable periods are critical to consider when thinking about potential power system extremes and the authors have published extensively on this topic in the past. We chose to focus on the summer extremes to present a potentially new system stress which could occur under climate change. However, to address this comment we have included an extra Figure, which includes histograms of GB wind power generation (onshore + offshore) at times of the coldest temperatures (using as a proxy for peak demand as demand itself is not produced here) to demonstrate the potential for wind power to provide assistance at times of peak demand. This is done for both the historical and future simulations to see how the magnitude of these extreme events is impacted by the climate adjustments. Text and Figures are pasted below which have been added to section 4.2

[Figure]

Figure 10: (left) The minimum 70 days of GB-average 2m temperatures over the historical period. (right) GB-average daily-mean wind power generation on the 70 days of lowest temperature for ERA5 (black) and the delta corrected ERA5 data (colours) See Table \ref{table1} for further details of the models used for the delta corrections.

Another relevant compound weather event is periods of low temperatures (associated with high demands \cite{Bloomfield2020}) and low wind speeds (resulting in low wind power generation). Figure \ref{coldest_days} shows histograms of the 70 coldest days from the ERA5 reanalysis and their corresponding wind power generation. The 70 days are chosen as this is equivalent to a once-yearly peak, although in reality these events will often be grouped together during the presence of high pressure systems (see \cite{Bloomfield2017}). An installed wind power capacity of 22GW is used, as taken from thewindpower.net database for 2021. Generally these cold temperatures are associated with relatively low wind power generation (with capacity factors of around 20$\%$. However, days of much higher generation are possible (with capacity factors up to 50$\%$). Figure \ref{coldest_days} also shows the potential impacts of climate change by including the 70 coldest days from each of the delta corrected simulations (coloured lines). In each simulations the climate is warming, resulting in potentially lower peak demands. However, there are not noticeable differences in the amount of wind power generation present. This shows that within our chosen climate scenarios wind power still has the potential to provide useful capacity at times of system stress.

* The python scripts in <https://researchdata.reading.ac.uk/331/> rely on a long deprecated python package `mpl_toolkits.basemap` which was deprecated in the favor of Cartopy. When including such a package it would be helpful to provide a general conda `environment.yaml` file or a pip `requirements.txt`. Otherwise it is hard for users to run the scripts.

This is a very good point, reference to this depreciated package has been removed and the code has been updated to be compatible with python3. We have also checked through all netCDF files after comments from the community to check all data is available and well documented within the README files.

**Technical Corrections**

* Equation (5) misses a closing parenthesis

This has now been included

**Comments**

Overall, the manuscript at hand is of high quality. The motivations and methodology are clearly described and the text is easy to follow.

The main purpose of the paper is to introduce datasets of bias-corrected historical and projected time series of weather and energy system variables at the European level. Historical data are based on ERA5. Future projections are based on selected global climate models. The datasets are suitably licensed under CC-BY 4.0 and accompanied by explanations in a README. NetCDF is the file format of choice.

I fully agree with the other reviewer (https://doi.org/10.5194/essd-2021-436-RC1). Rather than repeating their arguments, I would like to second all points raised to give them more weight.

Thank you for this, see the response to R1 for more responses to their comments, we have tried to address every point.

In particular, I would encourage the authors to evaluate whether

- they could extend the datasets to NUTS1 and NUTS2 across all of Europe within a reasonable amount of work to maximise the dataset's impact,

We agree that this would be very useful however this is beyond the scope of the project (which was set out to be very UK focussed) to create this data for the whole of Europe. This is why we pushed to provide the python scripts within the data repository so that others would be able to create similar outputs for their region of interest. Hopefully has part of future projects this dataset can be created for the whole of Europe.

- they could clear out deprecations in code and provide a dependencies file to execute the Python scripts in order to ease reproducibility,

Yes we agree this is very important and have made sure that the python scripts do not include any depreciated libraries.

- they could specify more prominently for which levels of radiative forcing the future time series are provided.

We have included information in section 3.1 on the climate scenario. For the PRIMAVERA project all data is created using the RCP 8.5 scenario. More information has been included in line 198.

\textcolor{red}{The rate of future warming in the climate model data used is taken as a multi-model mean from the RCP8.5 climate simulations from the 5th coupled climate model inter-comparison (see \cite{haarsma2016} for further details of the HighResMIP simulations). }

**Technical Corrections**

- line 336: 70 yer -> 70 year?

This has been corrected.